# The Paternal Experience of Fear of Childbirth: An Integrative Review

**DOI:** 10.3390/ijerph18031231

**Published:** 2021-01-29

**Authors:** Emma Moran, Carmel Bradshaw, Teresa Tuohy, Maria Noonan

**Affiliations:** 1St. Patrick’s Mental Health Services, D08K7YW Dublin, Ireland; 2Department of Nursing and Midwifery, University of Limerick, V94X5K6 Limerick, Ireland; carmel.bradshaw@ul.ie (C.B.); Teresa.G.Tuohy@ul.ie (T.T.); maria.noonan@ul.ie (M.N.)

**Keywords:** fear of childbirth, fathers, perinatal mental health

## Abstract

Background: It is estimated that approximately 13% of expectant fathers experience a pathological and debilitating fear of childbirth. Objective: The aim of this integrative review was to examine and synthesise the current body of research relating to paternal experience of fear of childbirth. Methods: A systematic literature search of five databases—CINAHL, Cochrane Library, MEDLINE, PsycArticles and PsycInfo—identified seventeen papers. Methodological quality of studies was assessed using the Crowe Critical Appraisal Tool. Results: Thematic data analysis identified three themes: the focus of fathers’ childbirth-related fears, the impact of fear of childbirth on health and wellbeing, and fear of childbirth as a private burden. Discussion: Fear of childbirth is a significant and distressing experience for expectant fathers who may benefit from an opportunity to express their childbirth-related fears in an environment where they feel validated and supported. Antenatal education is recommended to enhance fathers’ childbirth-related self-efficacy to reduce fear of childbirth. Conclusions: Fear of childbirth may negatively impact the lives of men and consequently their families. Further investigation into methods and models for identifying and supporting men at risk of or experiencing fear of childbirth is required to improve outcomes for this population of men.

## 1. Introduction

Fathers’ attendance at childbirth has changed over the past 40 years, with approximately 96% of fathers in the developed world now being present during birth [1]. This change has coincided with an evolution in the perceived role of the father. Twenty-first-century fathers are viewed of as being providers and protectors who take a proactive hands-on role with their children in addition to giving practical and emotional support to their partner [2]. Prospective fathers are relied upon by their partners for support and assistance during pregnancy, childbirth and in the raising of their children [3]. The modern gender constructs of ‘maleness’ suggest that men should be strong and self-confident, which does not encourage fathers-to-be to express fears about their own capabilities [4]. Men may question the legitimacy of their psychological experiences during the perinatal period as they view themselves in a supportive role to their partner and consequently are reluctant to express and communicate their fears around birth and fatherhood [4]. Fatherhood, even when it is desired and planned for, can be a difficult time of transition for some men, negatively impacting their mental health, resulting in stress, anxiety and depression [5,6].

Fathers experience of childbirth has been particularly understudied despite evidence suggesting that the experience of birth contributes to a fathers’ adjustment to fatherhood during the postpartum period [7]. When childbirth is experienced as a traumatic event by parents, this can negatively impact their postnatal mental health [8]. Childbirth can be perceived as a traumatic event by one or both parents when there is an actual or threatened risk of serious injury or death to mother or child [8]. This may result in some parents’ experiencing intense fear, helplessness and a sense of loss of control [9]. Experiencing a difficult or stressful birth can heighten any pre-existing parental emotional vulnerabilities, which can subsequently contribute to negative postpartum outcomes, including difficulties with bonding [7].

The incidence of paternal perinatal depression and anxiety is assessed to be approximately 5–10% and 5–15% respectively [2,10].

A pathological fear of childbirth (FOC) is estimated to effect approximately 13% of fathers-to-be [11]. However, the focus of most research is on maternal FOC and tokophobia. Limited research has explored paternal FOC using validated screening tools and there is a lack of consistent definition of paternal FOC [12] across the literature, making the true incidence rate difficult to determine. These fathers-to-be with a pathological FOC experience severe anxiety, which impacts their daily functioning, causes distressing feelings of helplessness and affects their ability to prepare psychologically for fatherhood [13] as opposed to the common worry and uncertainty experienced by fathers-to-be in anticipation of childbirth. Fathers are categorised as having mild, moderate or severe/intense FOC using screening tools such as the Fear of Birth Scale (FOBS), which determines cut-off points between these categories [4,14]. The most frequently used tool for identifying FOC is the Wijma Delivery Expectancy/Experience Questionnaire (W-DEQ). This 33-item Swedish measure was developed by Wijma, Wijma, and Zar [15] and is considered a validated screening tool for evaluating FOC among women, with scores ranging from 0 to 165. The W-DEQ screening tool was intended for use in women but it is frequently used in the literature to screen for FOC among men [13,16]. A number of studies have revised the tool for use among men by omitting questions perceived to be irrelevant to males [13,17]. However, the reliability and validity of this screening tool among men requires further investigation.

### 1.1. Fear of Childbirth and the Request for Caesarean Section

FOC is a significant obstetric concern as it often leads couples to request birth by caesarean section (CS), which they perceive to be a safer and more controlled birthing option [18,19]. CS rates performed without medical indication are rising worldwide, with an increasing tendency to perform CS on a mother’s request [20]. Maternal request for CS birth, commonly attributed to FOC, are estimated to account for approximately 7–22% of CS births in Finland, Sweden and the UK [21]. Similarly, Gao et al. [22], in a more recent study, estimated the rate of CS on maternal request as being 10–20% in Northern Europe, the USA, Sweden and Australia. However, it is unclear whether a fathers’ experience of FOC influences his partners request for caesarean birth, which warrants further exploration. The World Health Organisation [23] recommends the use of non-clinical interventions such as health education as a means to reduce the rate of unnecessary CS births [24].

### 1.2. Rational for Undertaking This Review

Pregnancy is a natural and often longed for progression in a couple’s life. It can, however, become a time of great worry and anxiety as preparations are made for the challenges that parenthood entails [25]. There is a wealth of research indicating that maternal stress is associated with adverse pregnancy outcomes [26,27]. Significantly fewer studies have examined the mental health challenges faced by fathers during this time [1,28]. A father’s involvement in the life of his child has consistently been shown to influence child outcomes independent of maternal factors including improved intellectual outcomes and a reduction in child behavioural problems [29,30]. A fathers’ experience of perinatal mental health conditions including anxiety and depression can impact on a fathers’ mental and physical health, personal relationships and parenting ability [31,32]. Therefore, conditions such as paternal FOC are in need of greater exploration and understanding. The identification and treatment of paternal perinatal mental health problems (PMHPs) are significant as they have a protective benefit to the entire family in addition to improving child outcomes [2].

The majority of research to date has assessed fathers’ anxiety during the perinatal period including a systematic review conducted by Philpott et al. [32]. This review found that factors contributing to fathers’ anxiety were lower education levels, lower household income, poor co-parenting support, a partner’s anxiety and depression and being present during a previous birth. However, there is a paucity of research examining fathers’ experiences of FOC. This review aims to identify, explore, critically appraise and synthesise the evidence relating to men’s experience of FOC in order to increase health professional’s understanding of this condition in men. A synthesis of the findings of the relevant research can identify knowledge gaps, areas for further research, provide guidance on preventative measures and inform the development of appropriate treatment interventions and, in so doing, enhance family centred care within the maternity services.

## 2. Methods

### 2.1. Design

An integrative review method allows for the simultaneous inclusion of various diverse study methodologies [33] and supports the collation, analysis and integration of findings from existing primary research relating to a specific research question [34]. An integrative review was identified as the most appropriate review method to critically appraise the literature evidence relating to fathers’ experience of FOC. This review was conducted in accordance with the Preferred Reporting Items for Systematic reviews and Meta-Analyses (PRISMA) [35].

### 2.2. Search Strategy

A search of the CINAHL, Cochrane Library, MEDLINE, PsycArticles and PsycInfo databases was conducted between 15 September 2019 and 15 January 2020 to identify primary studies related to the review question [36] using the following search terms: tokophobia OR tocophobia OR fear of childbirth OR childbirth fear AND fathers OR men OR dads OR paternal OR male. No restrictions were placed on publication date of studies as no previous systematic or integrative review had been identified on this topic, therefore warranting a full evaluation of available research. Details of inclusion/exclusion criteria are available in Table 1.

### 2.3. Search Results

A total of 297 citations were retrieved from selected databases and 201 papers remained after removal of duplicates. Screening of title and abstract excluded 160 articles which did not meet the inclusion criteria for the review, resulting in full-text review of 41 studies. A further 24 papers were excluded following full-text review as their content was not relevant to the review question (Figure 1). A total of 17 papers met the inclusion criteria for this review, consisting of four qualitative studies, eleven quantitative studies and two mixed-methods studies.

### 2.4. Critical Appraisal

Eligible studies were critically appraised using Crowe’s Critical Appraisal Tool (CCAT) [37]. CCAT scores among the studies varied between 19 and 32 out of a possible score of 40. The 17 papers were included in the review following quality appraisal, although a number of methodological limitations were identified as detailed in the data extraction table. Overall, the studies presented clear research aims, objectives and rationale for addressing their research question.

### 2.5. Data Extraction, Reduction and Analysis

Data from each of the 17 eligible studies were organised into a data extraction table (Table 2). The extraction fields chosen in this review were: author(s), year, title, country, study design/setting, sampling strategy/size, data collection methods, analytical approach, findings, strengths and limitations and quality appraisal. Data relevant to the review question was extracted from each selected study and analysed according to the thematic analysis (T.A.) approach described by Braun and Clarke [38,39].

Excerpts from the data findings section of each of the selected studies was transferred verbatim into a thematic synthesis table (Appendix A). Coded data developed through iterative and interactive readings of primary study findings was compared, contrasted and categorised into three descriptive themes agreed by authors (E.M., M.N.). A thematic map was created (see Figure 2) to outline themes and subthemes that emanated from the synthesis of data across eligible studies.

## 3. Results

### 3.1. Study Characteristics

A variety of research designs were represented among the studies in this review. This included four qualitative studies [1,40,41,42], eleven quantitative studies [4,11,13,14,17,43,44,45,46,47,48] and two mixed-methods studies [9,16]. Ten of the quantitative research designs were observational [4,9,11,14,43,44,45,46,47,48]. Qualitative study designs consisted of an ethnographic approach [42], a phenomenological design [1], and a grounded-theory design [40]. The design used by two of the studies was not clearly reported [9,41]. Hunter et al.’s [16] mixed-methods study employed an experimental repeated measures design with interview. Secondary analysis of results from a randomised control trial were reported in two studies [13,17]. The majority of studies were conducted in Sweden (*n* = 9) [4,11,13,14,17,40,43,45,48]. The remaining studies were conducted in the USA (*n* = 1) [16], South Africa (*n* = 1) [47], Northern Ireland (*n* = 1) [41], Israel (*n* = 1) [1], Hungary (*n* = 1) [44], Germany (*n* = 1) [46] and the United Kingdom (UK) (*n* = 2) [9,42]. Sample sizes ranged from 8 to 20 participants among the qualitative studies, 9 to 11 participants in the mixed-methods studies and between 150 and 1105 in the quantitative studies. Nine of the seventeen studies included a mixed sample of both first-time fathers and fathers with one or more children [4,11,13,14,40,43,45,46,48]. Six studies sampled first-time fathers only [1,9,16,17,42,47], and two studies did not provide details of the fathers’ previous children [41,44].

Two studies [13,17] present secondary analysis of data collected from a randomised control trial [49] designed to investigate two models of antenatal education. Three papers reported on different results from the same longitudinal cohort study [14,43,45] and two papers reported on different findings from the same cross-sectional survey [4,11]. This culminated in inclusion of a total of 17 papers, representing 13 primary studies. Various scales were used across the studies including the W-DEQ [13,16,17], the FOBS [43,45], the Cambridge Worry Scale [13,17], and the Impact of Event Scale [9], and several researchers developed self-rated questionnaires [4,11,40,45,46,47,48]. Many of the studies tested the reliability of their chosen instruments for use in fathers using split-half reliability; Cronbach’s alpha and construct validity [16]; Cronbach’s alpha [43]; factor analysis [11]; pilot testing [17,47]; pilot testing and Pearson correlation coefficient [13]; analysis of variance [44]; and face-to-face validation [48]. Authors of three studies did not detail validity measures [14,40,45].

**Table 2 ijerph-18-01231-t002:** Data extraction table.

Author(s), Year, Title, Country	Study Design and Setting	Sample Size and Strategy	Data Collection Methods	Analytical Approach	Findings	Strengths/ Limitations	Quality Appraisal Including CCAT Score
Bergström et al. (2013) [13] Fear of childbirth in expectant fathers, subsequent childbirth experience and impact of antenatal education: sub analysis of results from a randomised controlled trial. Sweden.	Quantitative sub analysis of results from a randomised controlled trial (RCT). Of the 83 men identified as having FOC, 39 were randomised to psych prophylaxis childbirth preparation training. The remaining 44 men received standard antenatal preparation without such training. 15 antenatal clinics in Sweden.	Convenience sample of 762 men, of whom 83 (10.9%) were identified as a subsample suffering from FOC based on the data from the W-DEQ.	The W-DEQ, the Cambridge Worry Scale (2 of 16 items excluded and 1 item altered for use in men).	Secondary statistical analysis from a RCT investigating two models of antenatal education. Chi-square tests, *t*-tests presented as means and standard deviations (SD).	Men with antenatal fear of childbirth more often experienced childbirth as frightening (OR 4.68, 95% CI) and reported feeling unprepared for childbirth (OR 4.04; 95% CI 2.08–7.84) compared with men without fear. Participants in the psych prophylaxis group had a lower risk of experiencing childbirth as frightening compared with those receiving standard antenatal preparation (OR 0.30: 95% CI 0.10–0.95).	The W-DEQ piloted for use in men prior to use and 8 of the 33 items were excluded since they were deemed irrelevant to men. Cut-off value of >60 set on the adapted W-DEQ. Further testing of the validity and reliability of the instrument for use in men is warranted. Secondary analysis of data from a trial originally designed for other hypotheses.	CCAT score 31/40 (77%). Validity of the W-DEQ tool tested through piloting. Conflict of interest and relationship between researchers and participants not addressed.
Chalmers and Meyer (1996) [47] What men say about pregnancy, birth and parenthood. South Africa.	Quantitative study (methodological design not identified). One public hospital and one private hospital.	Convenience sample of 150 first-time fathers split into three groups of 50. Participants were recruited equally between the two hospital settings.	Fathers in each group were asked to complete one questionnaire on their perception of their partners’ pregnancy (response rate 92%, *n* = 46), or their experience of antenatal education (response rate 72%, *n* = 36), or their experience of the birth (response rate 66%, *n* = 33). A follow-up questionnaire about parenthood experience was requested of all participants, with a response rate of 49.6% (*n* = 57).	Descriptive statistics.	The most significant fears experienced by men during pregnancy were the fear of abnormality in the baby (71%), not being at the birth (47.8%), partner experiencing pain (43.5%), and partner or baby dying (41.3%). 30.4% of men reported feeling more anxious than before pregnancy, more emotional (13.4%) and more irritable (8.7%). The most important source of emotional support for men was their partner (63.9%).	Methodological and data analysis approaches not identified. Cultural bias identified. Screening tools not used to assess FOC. Participants recruited from both the public and private hospital to represent both health care systems but no comparative results reported between the two hospitals.	CCAT score 19/40 (47%). Methodological design not explicit. Sample bias identified. Reports that questionnaires were pilot tested but no details provided of pilot.
Eriksson et al. (2005) [11] Experiential factors associated with childbirth-related fear in Swedish women and men: A population based study. Sweden.	Quantitative cross-sectional survey. Swedish hospital setting.	A convenience sample of 558 women and 552 men. Response rate (*n* = 410, 73% women) and (*n* = 329, 59% men). Participants identified via health records of their infants at primary health care centres.	Questionnaire, pilot tested for face validity.	Descriptive statistics including means and standard deviations.	13% of the men were assessed as having intense fear of childbirth, 29% moderate fear. 98% of men with intense fear and 92% with mild/moderate fear felt afraid that their child would not be born healthy. Men with intense fear were more often 40 years of age or older. 56% of men with intense fear did not disclose their fear as they did not want to worry their partner. 49% with intense fear felt it best to keep the fear to themselves.	Lower response rate among male participants. Questionnaire was pilot tested in 10 men and 10 women, with minor adjustments made prior to use in study.	CCAT score 31/40 (77%). Possible response bias identified due to lack of information on non-responding male participants. Retrospective design may have introduced recall bias. Conflict of interest not reported.
Eriksson et al. (2006) [4] Content of childbirth-related fear in Swedish women and men- analysis of an open-ended question. Sweden.	Quantitative Cross-sectional survey, retrospective design. Hospital setting in Northern Sweden.	A random sample of 558 mothers and 552 fathers. Response rate to questionnaire (*n* = 410, 75%) mothers and (*n* = 329 60%) fathers. Of the respondents with experience of childbirth-related fear (*n* = 308, 94%) mothers and (*n* = 194, 82%) fathers answered the open-ended question. Participants identified via health records of their infants at primary health care centres.	Self-rated questionnaire. Use of both fixed and open-ended questions.	Open-ended question analysed using content analysis. Chi-square tests were used to report proportional differences between participants with intense and mild–moderate FOC.	Predominant fears of fathers included the health and life of the baby (79%), injury to the child during birth (41%), the health and life of the woman (49%), the woman being injured during labour (45%).	Questionnaire was answered 1.5 years after the birth (retrospective design). The time delay may have altered recall of the birth event and/or specific details. Participants did not consent to have their details taken from health records. Open-ended questions enabled parents to describe their fears in their own words, giving a deeper insight. Content analysis coded by all three researchers.	CCAT score 28/40 (70%). Recall bias.
Eriksson et al. (2007) [40] Men’s experience of intense fear related to childbirth investigated in a Swedish qualitative study. Sweden.	Qualitative study, grounded theory design Interviews conducted in a setting of the participants’ choice.	Sample of 22 men. Sampling strategy not identified. Participants were identified as having FOC through participation in a previous survey.	Interviews guided by open-ended questions and a permissive strategy.	Similarity- difference grounded theory approach (Strauss and Corbin, 1990).	Content of childbirth fear was primarily described as being related to the health and life of their partner and child, obstetric staff competence/behaviour and their own capabilities /reactions. The manifestation of fear was often described as a mental occupation. Some of the strategies that participants used to deal with their fear was an attempt to increase their sense of control and diminish the emotion of fear.	All 3 authors participated individually and collectively in coding and characterisation of data and in establishing meaning and content, which added to the confirmability and reliability of reporting. Validity of original survey tool for assessing childbirth fear among participants unclear.	CCAT score 32/40 (80%). Fathers’ interviews were conducted between two and three years after their child’s birth, which may have introduced a recall bias.
Etheridge and Slade (2017) [9] “Nothing’s actually happened to me.”: the experience of fathers who found childbirth traumatic. U.K.	Mixed-methods study.	A volunteer sample of 11 fathers. Participants recruited via advert on the Birth Trauma Association Website, in a newsletter and on two internet forums.	The Impact of Event Scale (IES) questionnaire and semi- structured telephone interviews.	Thematic analysis using template analysis.	10 of the 11 men (90%) described fears that their partner or baby would die. The pain of the woman and her suffering had a direct effect on the man and his distress mirrored hers. 7 fathers (63%) referred to “trying to keep it together” and be strong for their partner. Preoccupation and rumination was a feature for some men in the weeks, months and even years after the birth.	Reliability and validity of IES reported. Two men had received previous treatment for depression. Variations in length of time since birth ranged from 2 months to 6 years, which may introduce recall bias.	CCAT score 32/40 (80%). Participant’s right to withdraw reported. Volunteer bias. Suitability of sampling method or sample size not reported.
Greer et al. (2014) [41] ‘Fear of childbirth’ and ways of coping for pregnant women and their partners during the birthing process: a salutogenic analysis. Northern Ireland.	Qualitative study (methodological underpinning not reported). Some of the interviews were conducted in the hospital setting and others in the participant’s homes.	A purposive sample of 19 women and 19 men.	In-depth semi- structured interviews.	Thematic content analysis.	Participants were fearful that their partner would be unable to cope with and be traumatised by the pain of childbirth and that their partner’s postnatal mental health would be affected. Participants felt labour and birth posed considerable risks to the physical health of the mother and baby. Some of the participants feared that their baby was too big to be born vaginally.	Study participants all attended the same health care setting (consultant-led hospital), which may have impacted the heterogeneity of sample. Confidentiality, privacy and informed written consent all considered.	CCAT score 26/40 (65%). No demographic details of participants reported but available through visiting a website. Dependability and rigor of data analysis method not reported. Conflict of interest not reported.
Hildingsson (2014a) [14] Swedish couples’ attitudes towards birth, childbirth fear and birth preferences and relation to mode of birth- A longitudinal cohort study. Sweden.	Quantitative Longitudinal cohort study. Three hospital settings Mid-North Sweden.	A convenience sample of 1074 pregnant women and their partners.	Two questionnaires—first one administered mid-pregnancy and the second two months after the birth.	Descriptive and inferential statistics (chi-square and *t*-tests) and multinomial regression analysis.	15% of women and 5% of men had childbirth fear. Birth preferences and fear were strongly associated with mode of birth. Men rated women’s health and wellbeing higher while women prioritised the baby’s health.	Not all dimensions of FOC were covered within the questionnaire as overall purpose was to explore various components including early parenthood. A validated tool to screen for FOC was not utilised.	CCAT score 31/40 (77%). Confounding variables reported. Suitability of sample size was not discussed.
Hildingsson et al. (2014b) [43] Childbirth fear in Swedish fathers is associated with parental stress as well as poor physical and mental health. Sweden.	Longitudinal regional survey. Three hospital settings Mid-North Sweden.	A convenience sample of 1047 expectant fathers. 59% (*n* = 620) of fathers completed all three questionnaires.	Three questionnaires—first delivered in late pregnancy, second at two months postpartum and third one year postpartum. The Fear of Birth Scale (FOCS), self-reported physical and mental health assessment, and the Swedish Parental Stress Questionnaire (SPSQ).	Descriptive statistics. Crude and adjusted odds ratios (OR) with a 95% confidence interval (CI) were calculated between fathers who scored higher fear of birth and those who scored lower fear of birth.	Childbirth-related fear was present in 13.6% of fathers as assessed using the FOCS. Respondents with scores of >50 in the FOCS were identified as those with greater fear. These fathers were more likely to rate their physical (OR 1.8; CI 95% 1.2–2.8) and mental health (OR 3.0; 1.8–5.1) as poor compared to fathers without FOC. Fearful fathers were more likely to perceive difficulties in pregnancy (OR 2.1; 1.4–3.0) forthcoming birth (OR 4.3; 2.9–6.3) and parenthood (OR 1.4; 0.9–2.0) than fathers without FOC. Higher levels of self-rated stress were also present in men with FOC at 12 months postpartum.	Limited to Swedish-speaking fathers only. Large sample size but a high level of non-responders (41%) for final stage. Convenience sample may cause volunteer bias.	CCAT score 28/40 (70%). Reliability and validity of the FOCS and cut-off point of >50 highlighted. Confidentiality and researcher’s relationship with participants not reported.
Hildingsson et al. (2014c) [45] Childbirth fear in expectant fathers: Findings from a regional Swedish cohort study. Sweden.	Regional cohort study, part of a prospective longitudinal cohort study. Three hospital settings Mid-North Sweden.	A convenience sample of 1414 expectant fathers. Response rate (*n* = 1047, 74%).	Self-reported questionnaire using a five-point Likert scale and the FOCS.	Statistical analysis using crude and adjusted odds ratios with a 95% CI, logistic regression analysis.	13.6% of expectant fathers were identified as having FOC through assessment using the FOCS. Scores of >50 were used as a cut-off point to identify those with FOC. Fathers reporting FOC were more likely born in a country outside Sweden (OR 2.8; 1.3–6.1) be first-time fathers (OR 1.8; 1.2–2.6), prefer a caesarean birth (OR 2.1; 1.7–4.1) and have more frequent childbirth-related thoughts in mid-pregnancy (OR 1.9; 1.1–2.0). Men with FOC were also less likely to agree with the statement that giving birth is a natural process.	Reliability and validity of the FOCS for use in men not reported. Fairly large sample size across three hospital settings.	CCAT score 30/40 (75%). Validity of cut-off point for the FOCS >50 not determined in male population. Low response rate.
Hunter et al. (2011) [16] Satisfaction and use of spiritually based mantram interventions for child-birth related fears in couples. U.S.A.	Mixed-methods design. Experimental and interviews. Urban military medical centre.	A convenience sample of 20 pregnant women and 9 male partners. Randomly assigned into intervention or control group. Control group: childbirth course only. Intervention group: childbirth course and mantram program.	The W-DEQ, the Client Satisfaction Questionnaire and six-month follow up via telephone interview.	Descriptive statistics and inferential statistics (t tests, chi-square and Cramer’s V statistics).	Males’ W-DEQ scores ranged from 73 to 96. No significant difference in FOC between intervention and control group. There was not sufficient evidence to confirm that mantram repetition is beneficial for managing FOC due to small sample size and incomplete data from questionnaires. Eight respondents in the intervention group (Women (*n* = 5) and men (*n* = 3)) completed a satisfaction questionnaire. 75% reported high satisfaction and 25% medium satisfaction. There was no breakdown of these percentages by sex.	The majority of participants were active duty military or military dependants, which may reduce generalisability. Convenience sample may lead to volunteer bias. Poor enrolment rate of 20% (of 134 potential participants) resulted in small sample size. No fathers completed the six-month follow-up interview. Uneven distribution of ethnicity.	CCAT score 28/40 (70%). Sampling bias identified. Construct validity was performed for use of the W-DEQ by means of correlation with other questionnaire scales.
Kannenberg et al. (2016) [46] Treatment-associated anxiety among pregnant women and their partners: What is the influence of sex, parity, age and education? Germany.	Quantitative cross-sectional survey. Women’s hospital setting, Germany.	A sample of 259 pregnant women and 183 male partners. Sampling strategy not identified.	State-Trait Anxiety Inventory (STAI) and self-assessment questionnaire.	Statistics ANOVA and *t*-tests.	Fathers did not report reduced scores for FOC with second or subsequent children as was found for women. Fear of foetal malformation was found to be more anxiety provoking in parents with higher levels of education. Fear for the unborn child’s health was the most prominent fear. Anxiety rose in both men and women as gestational age increased.	Reliability and validity of questionnaires used not reported. Methods to ensure participant’s confidentiality not reported. Conflict of interest reported.	CCAT score 24/40 (60%). Study participants consisted of couples attending hospital care who are at higher risk of obstetric complication than those who attend practice gynaecologists or midwifery care in this health setting. Thus results may not be fully representative.
Schytt and Hildingsson (2011) [48] Physical and emotional self-rated health among Swedish women and men during pregnancy and the first year of parenthood. Sweden.	Quantitative longitudinal study. Three hospital settings Mid-North Sweden.	Sample of 1506 women and 1414 male partners. Response rate: 80% (*n*= 1212) women and 78% (*n*= 1105) men completed the first questionnaire, 50% (*n*= 763) women and 46% (*n* = 655) men completed the final questionnaire.	A total of 4 questionnaires—Q1: completed in the second trimester, Q2: in the third trimester, Q3: two months postpartum and Q4: one year postpartum.	Statistical analysis (Friedman’s test, Wilcoxon signed rank test).	30% of men with childbirth-related fears rated poor physical self-rated health and 27% poor emotional self-rated health in late pregnancy. With poor physical self-rated health among 42% and poor emotional self-rated health in 37% one year after birth. Poor emotional self-rated health was associated with having children previously, childbirth-related fear, pronounced emotional changes during pregnancy and perceived stress when facing the forthcoming parenthood.	Reliability and validity of questionnaire assessing childbirth-related fears not reported. Ethical approval and any conflict of interest not reported.	CCAT score 27/40 (67%). High attrition rates. Participation was limited to those with mastery of the Swedish language.
Schytt and Bergström (2014) [17] First-time fathers’ expectation and experience of childbirth in relation to age. Sweden.	Secondary data analysis from a randomised control trial. 15 antenatal clinics across Sweden.	Of the 1064 trial participants, 777 first-time fathers who completed the follow-up questionnaire were included. Divided into three groups: young men <27 years (*n* = 188), men of average age 28–33 years (*n* = 389) and men of advanced age >34 years (*n* = 200).	Two questionnaires—first completed in mid-pregnancy the second at follow-up 3 months postpartum. The W-DEQ was used to measure fearful expectations. Single-item questions on worry were retrieved from the Cambridge Worry Scale.	Statistical analysis (X^2^-tests, *t*-tests, multivariable logistic regression analysis).	29% of the advanced aged men reported mixed or negative feelings compared with 27% average age and 17% young age (*p* = <0.05). Fearful expectations were most pronounced in the older cohort of men. The total sum score on the W-DEQ for men in the advanced age category was 43.3 (SD 16.9), compared with 42.9 (SD 13.5) in men of average age and 38.7 (SD 15.7) in the youngest age category.	The W-DEQ piloted for use in expectant fathers and validity of scale reported. Large sample size.	CCAT score 32/40 (80%). Sample bias may exist as participants took part in a trial on antenatal education and may not be representative of population. Confounding variables reported.
Shibli-Kometiani and Brown (2012) [1] Fathers’ experiences accompanying labour and birth. Israel.	A phenomenological qualitative study. Interviews took place in participants’ own home.	A purposive sample of 8 fathers. Suitable participants were identified through the labour ward register.	Semi-structured interviews.	Colaizzi (1978) framework for data analysis.	Every participant expressed significant levels of fear, anxiety and helplessness as labour progressed. They feared their partners and baby might die. As their distress increased, they became passive and less supportive. A knowledge deficit about labour served to increase their anxiety.	Couples the researcher had cared for personally were excluded to reduce bias. Management of data, confidentiality, written consent and participants’ right to withdraw were reported. Sample representative of the cultural diversity of the region. Small sample size. Screening tools not used to assess FOC.	CCAT score 26/40 (65%). Limited discussion and interpretation of results within context of current knowledge. Rigor of chosen analytical method not reported. No statement of ethical approval.
Somers-Smith (1998) [42] A place for the partner? Expectations and experiences of support during childbirth. U.K.	A qualitative study using an ethnographic approach Two consultant-led maternity units in Hampshire, U.K.	A purposive sample of 13 couples; response rate 61% (*n* = 8 couples).	Two semi-structured interviews—first conducted six weeks before the birth and second approx. twelve weeks after the birth.	Thematic analysis guided by Miles and Huberman (1994).	One fear voiced was the possibility of their partner dying. Other fears men expressed were the possibility of fainting, panicking and if they would be able ‘to keep it together’. The men mostly kept their fears to themselves. One father relied on cues from the midwife to minimise his anxiety during the labour.	Small sample with refusal rate of 39%. No participants from lower social economic groups and limited ethnic diversity among participants. Screening tools not used to assess FOC.	CCAT score 28/40 (70%). Exclusion criteria not reported. Relationship between the researcher and participants not reported. Conflict of interest not reported.
Szeverényi et al. (1998) [44] Contents of childbirth-related fear among couples wishing the partner’s presence at delivery. Hungary.	Cross-sectional survey. Distributed at self-referred antenatal parent craft preparation course.	Convenience sample of 216 couples.	Questionnaire designed by Ringler (1985) included 49 items for women and 52 items for men.	Analytical approach not identified. Statistical summary of results presented in tables.	Approx. 80% of couples had fears relating to childbirth. 13% of men had a strong fear and 11% a very strong fear of caesarean delivery. 15.7% of men were very afraid that their wife could die and 5.6% quite afraid. 14.8% were very afraid their baby may be stillborn and 11.1% quite afraid.	First study of its kind in Hungary. Only partners who attended the 3 parent craft preparation courses were permitted to attend the birth of the baby. 100% of couples completed the questionnaire. Potential response bias and lack of generalisability may be present due to the couples being self-referred. Screening tools not used to assess FOC.	CCAT 19/40 (47%). Poorly described methodology. Validity and reliability of questionnaire reported using analysis of variance. Suitability of sample size and exclusion criteria not reported. Ethical approval not reported.

W-DEQ—Wijma Delivery Expectancy/Experience Questionnaire; CCAT—Crowe’s Critical Appraisal Tool [37].

### 3.2. Focus of Fear

The first theme that emerged from a synthesis of findings across studies was the focus of fear, which offers an insight into what men with FOC are fearful of in relation to childbirth [1,4,9,11,40,41,42,44,46,47] and is explored under the following subthemes: the health and life of the baby, the health and life of their partner, and reactions and behaviour.

#### 3.2.1. The Health and Life of the Baby

The most commonly expressed paternal fear was for the health and life of the baby [1,4,9,11,40,44,46,47]. Findings from Eriksson et al.‘s [11] study found that 98% of men with intense childbirth fear and 92% with mild–moderate fear believed that every expectant parent is afraid that their baby would not be born healthy. Specific fears that the baby would be born with an abnormality, disease or “handicapped” were reported by participants in two studies [4,47]. In Chalmers and Meyer’s study [47], 50% of the responding fathers reported checking the baby for an abnormality at birth. Fear of malformation in the baby was found to be more anxiety provoking among parents with higher education levels [46]. The baby experiencing a birth injury was a very strong fear among 13% of fathers surveyed by Szeverényi et al. [44]. In many of the studies, a significant and predominant fear was the fear that their baby would be stillborn or might die [1,4,9,42,44,47]. Men’s fear levels did not reduce with increasing parity, in contrast to fear levels in parous women which were very often reduced [46].

#### 3.2.2. The Health and Life of the Partner

Along with the fear that their baby may die, fathers also expressed significant fears for the health of their partners [4,40,41,45,47]. Fathers were found more likely to fear for the welfare of the woman than the women themselves [4]. Greer et al. [41] found that many fathers (42%) feared that their baby was too large to be born vaginally and their partner would be unable to cope with the pain. Similar results were found by Chalmers and Meyer [47], where 43.5% of fathers feared their partner experiencing pain. Other fears reported across the studies included injury to their partner [4], partner’s postnatal mental health being negatively affected by a traumatic birth [41], prolonged birth [4], episiotomy [47], fear of interventions [4] and caesarean birth [44]. Fears relating to the health and life of the woman and the health and life of the baby were reported significantly more often by men with intense FOC than by men with a mild or moderate fear [4].

One study found that men were more positive than women about the use of medical interventions during labour, such as the use of epidural [41]. All of these men wanted their partner to have as much pain relief as possible during childbirth in the hope that it would make the birth easier and safer. Similar findings were reported by Hildingsson [14] who found that men desired a birth that was the safest, least painful and stressful option for the woman and one that optimised recovery. A perception that childbirth is intrinsically dangerous was increased when there was history of a previous negative birth experience [41].

#### 3.2.3. Reactions and Behaviour

A number of studies found that fathers experience fears relating to their own capabilities and reactions, as well as concerns regarding the competence and behaviour of health professionals [4,40]. Men were fearful of doing something wrong during labour [44], and doubts about their own reactions were common [40]. Other reported fears were not being able to give help and support to their partner during childbirth and not being able to endure the situation [4]. A lack of knowledge about childbirth served to increase fathers’ anxiety [1]. Fathers expressed fear that they may not be present at the birth [47], and the possibility of fainting or panicking [42]. Fathers with FOC were found to feel more unprepared for childbirth and experience childbirth as a more frightening experience than fathers who did not report fear [13].

### 3.3. Impact on Health and Wellbeing

The second theme reports on the impact FOC has on fathers’ health and wellbeing under three subthemes: mental health, physical health, and coping mechanisms and avoidance.

#### 3.3.1. Mental Health

The impact of experiencing FOC on fathers’ mental health was reported across twelve studies [1,9,11,13,17,40,41,42,43,45,47,48]. Some fathers described fear of childbirth in terms of a mental preoccupation in which they thought about their fear to a great extent [40]. Other participants had not been as consumed by fear but reported that the fears brought with it a sense of increased vigilance. Fathers with FOC were found to be twice as likely to report they were thinking about the birth of their baby compared to those without FOC [45].

In one study, men reported preoccupation, rumination and flashbacks in the weeks, months and years following the birth [9]. This study also found that the pain and suffering of the woman in labour had a direct effect on the man, with his distress mirroring hers [9]. Shibli-Kometiani and Brown [1] reported that as labour progressed, men’s anxiety, fear and distress increased and with it they became passive and less supportive. Despite their fear constituting emotional distress, a number of more positive aspects emanating from their fear were reported by fathers in one study [40]. One man reported that his fear had helped to make the pregnancy and impending birth seem more real and another man described his fear as giving him greater insight into what is really important in life.

#### 3.3.2. Physical Health

Physical health symptoms experienced by men with FOC were reported across five studies [9,40,42,43,47]. Experiencing FOC impacted fathers’ physical health in various ways including changes in perception and a sense of heightened physical awareness [9], sleep disturbance, weight gain, nausea [47], and uneasy body sensations, restlessness, stomach ache and erectile problems [40].

#### 3.3.3. Coping Mechanisms and Avoidance

Coping mechanisms utilised by some of the fathers were identified across studies [9,16,40,43]. Fathers attempted to increase their sense of control in a number of ways including preparing for the birth, relieving the woman of her daily chores and restricting the woman from pursuing any activities they deemed harmful [40]. Others engaged in more frequent religious practice [16], and kept busy with extra work and physical training [40]. Some fathers avoided external stimuli such as television programs on childbirth which triggered memories of birth [9]. Other men with FOC avoided attending antenatal care visits as they did not want to talk about everything that could happen [40]. However, these results are not supported by Hildingsson et al. [43], who found that fathers with previous children and FOC attended antenatal visits to the midwife more often than fathers without fear.

### 3.4. A Private Burden

This third identified theme is examined under the subthemes of gender constructs, non-disclosure and giving and receiving support. It captures a sense of what the experience is like for men with varying levels of FOC.

#### 3.4.1. Gender Constructs

All of the men in the Eriksson et al. [40] study felt that talking about childbirth-related fears was difficult and was not in the nature of men. Participating fathers referred to societal expectations and their responsibility to be strong and look after the woman’s needs. Fathers expressed the opinion that it did not seem appropriate to talk about their own fears, while others spoke about not wanting to look weak. Similarly, fathers in the Etheridge and Slade [9] study referred to trying to keep strong for their partner which was motivated by the thought that seeing him upset was not going to do her any good.

#### 3.4.2. Non-Disclosure

Men’s experience of childbirth fear often became a hidden and personal burden. A number of studies identified that most of the fathers had not expressed or spoken about their fears to either their partner, friends or relatives [40,42]. Fathers did not want to worry their partner and were afraid that talking about their fears might generate fear in the woman [11]. Other reasons given for not expressing their fear were not wanting to disappoint their partner or give her the impression that she could not expect his support, feeling the issue was of no interest to anybody and that talking would make their fears worse, suggesting a sense of isolation some men feel. On reflection, some men felt that talking about their fear may have been beneficial and even desirable [40].

Etheridge and Slade [9] found that during childbirth, most men tried to hide their feelings but became so overwhelmed that they broke down. The men who did manage to contain their emotions during the birth reported becoming extremely distressed and breaking down in tears when on their own, describing feelings of utter helplessness. Fathers were afraid that others may not understand or would dismiss their distress, which affected how they shared their experiences of birth [9]. Fathers in this study felt that they did not have the right to be affected because “nothing actually happened to me”. Eriksson et al. [11] found that 65% of men with intense fear reported that nobody ever asked them how they felt about childbirth, suggesting that lack of assessment and validation of men’s childbirth-related fears may be contributing to their lack of disclosure.

#### 3.4.3. Giving Support

Men were found to experience giving support during childbirth as being harder than they had expected and expressed concerns in relation to their ability to meet the needs of their partner [1,41,42]. Feelings of uncertainty and perceiving themselves as unable to do anything practical increased fathers’ feelings of helplessness [9,42]. Notably, a minority of participants in one study expressed more confidence in their ability to provide support to their partner following a CS birth rather than support during active labour [41]. Overall, men felt most comfortable with a practical and well-defined supportive role [1,42].

#### 3.4.4. Receiving Support

Fathers’ feelings about receiving support was reported in five studies [1,9,41,43,47]. Most of the men expressed a need for support [1]. The partner was identified as the most important source of emotional support for fathers [47]. Other sources of support were midwives [41,43]. In contrast, one study found that putting their trust in health professionals gave fathers a feeling of unease as this was in contrast to their usual experience of feeling in control [9].

## 4. Discussion

The aim of this review was to identify, explore, critically appraise and synthesise the evidence relating to men’s experience of FOC. A synthesis of findings across the selected studies identified the nature of paternal childbirth fears, the impact of FOC on paternal perinatal wellbeing, the silence surrounding this fear and the coping strategies that fathers engaged with to respond to their fears.

### 4.1. Paternal PMH and Childbirth Fears

Increased awareness of fathers’ perinatal mental health needs in relation to FOC, including risk factor and symptom identification, and the development and provision of support structures may reduce the negative impacts of FOC on postnatal mental health difficulties and improve outcomes for men, women and their families [50,51,52]. Early identification of fathers with FOC would support the planning of timely interventions to reduce the distress and potential negative impacts for fathers with severe FOC [52]. However, findings of this review offer only a limited examination of the factors associated with men’s risk of developing FOC outside of broad demographic explanations such as increased risks in first-time fathers [13,45], inconsistent findings re fathers’ age [11,17,46], the father’s country of birth [45], frequency of religious practice [16], and educational factors [46].

Effective screening methods are central to identifying fathers with PMH needs. However, despite FOC being a serious perinatal mental health condition there is a lack of clear recommendations regarding assessment tools and appropriate timing of screening [53]. There is a need for reliable and valid screening tools to identify FOC in fathers. However, screening in pregnancy is insufficient to improve clinical outcomes in fathers without appropriate follow up for diagnoses and treatment. The establishment of multidisciplinary care pathways for the treatment of fathers’ perinatal mental health conditions including FOC would facilitate staff confidence and encourage enquiry about fathers’ mental health needs [54].

### 4.2. Overcoming Barriers and Screening

This review identifies a number of barriers faced by those delivering antenatal care to identify men experiencing FOC. Non-disclosure is one barrier evident across studies [9,11,40,42]. Fathers can be reluctant to express their FOC support needs or seek help for fear that in doing so would detract from their partner’s needs [2,55]. Fathers want professional support from midwives but not at the expense of the mothers’ needs [56]. It is widely acknowledged that men’s general reluctance to talk about their FOC may lead to an underestimation of the problem [13,40,57]. Men find it difficult to seek help for mental health problems in general, are more likely to express negative attitudes toward therapy, and are more likely to discontinue treatment than women [58]. Professionals need to be proactive and flexible in support provision, which would limit the impact of personal barriers and encourage fathers’ active engagement [50]. Offering fathers with FOC their own appointment with the midwife would give the man an opportunity to talk about his fears without the risk of causing fear to his partner [40].

Men’s difficulty in speaking about their FOC may be related to socially constructed norms and expectations [40]. A systematic review by Ruffell et al. [50] found feeling overwhelmed, worrying they were a burden to others, and believing that others would see them as ‘weak’, were significant challenges for men in seeking support. A key challenge for health professionals is to be mindful of the unconscious gendered stereotypes that may exist as a result of socialisation concerning gender and parental roles [55]. Seeking therapy conflicts with masculine norms of self-reliance and can inhibit fathers seeking help and successfully engaging in treatment [58]. Recent evidence highlights the need to develop suitable gender-oriented PMH interventions for men which are sensitive to a father’s role [31]. A comprehensive knowledge of paternal PMH needs in health care professionals can help to break down barriers and ensure appropriate referral [59].

### 4.3. Self-Efficacy and Antenatal Education

The results of this review suggest that fathers with FOC often have a poor sense of self-efficacy in relation to childbirth. Increasing childbirth-related self-efficacy through interventions such as antenatal education may improve parents’ sense of control and confidence [60,61,62]. Despite the intention to involve fathers during the antenatal period, fathers are often described as having a secondary role during antenatal care and education [63]. Men have described feeling marginalised, invisible, side-lined and ignored during their experience of antenatal care with their partner in what they described as a female-dominated arena [64,65].

This review identifies that some men with FOC are found to avoid antenatal education in an attempt to not increase their fears [40], while, in contrast, other fathers with FOC were found to engage in more frequent antenatal visits compared to fathers without FOC [43]. Calls have been made for antenatal classes specifically for men and that address issues of concern for men as fathers may feel more at ease expressing their fears among an all-male group [2,66,67,68]. Further research is required to evaluate the effect of co-designed antenatal education specifically for men with FOC [62].

The findings of this review suggest that men underestimate their supportive capabilities and the comfort they are providing their partners. Therefore, health professionals have an important role in engaging fathers in antenatal conversations about their birth expectations, feelings, fears and role in childbirth to alleviate anxiety, increase self-confidence and promote relaxation [12,69]. Effective communication between the mother, father and midwife can make a difference to the level of control and connection that fathers feel at birth, which in turn influences their development of positive or negative birth perceptions [70].

Childbirth can be a time of psychological distress for men with FOC as they attempt to maintain support for their partner whilst managing their own anxiety [9]. Providing fathers with clear communication on the health of their baby and partner, can help alleviate the expectant father’s anxiety. A review by Hanson et al. [3] identified that fathers seek reassurance that they are doing the right thing for their partner during childbirth. Positive involvement of men in the perinatal period has the potential to decrease men’s anxiety, increase their trust in health professionals and offers them the opportunity to engage in psycho education [12]. Each of these elements helps to strengthen their role as they transition into fatherhood, and ultimately helps to shape the health and wellbeing of the new family. Findings from Greer et al. [41] suggest that some men feel more confident in their ability to support their partner after a CS birth, supporting the argument that men desire a more defined and practical role. In a review of fathers’ experiences of childbirth, Dellmann [71] found that providing practical and emotional support to their partner during childbirth alleviated men’s feelings of helplessness, making them feel useful and appreciated.

The specific focus of fathers’ childbirth-related fears was the most dominant and frequently reported finding across selected studies. Fathers fear most for the health and safety of both their partner and infant during childbirth and this finding is reported consistently across the broader literature [12,21]. Results from three of the review studies suggest that men with FOC perceive childbirth to be risk laden and are more likely to favour the use of medical intervention during labour than men without FOC [11,14,41]. Men are particularly fearful of the risks of childbirth when there is a history of a previous negative birth experience [41]. This is supported by evidence from a large Swedish study of 1105 expectant fathers which found that fathers with a previous negative birth experience were less likely to agree with the statement ‘Giving birth is a natural process’ and more likely to express a strong preference for birth by CS [72]. This finding needs to be taken into consideration when supporting women requesting a caesarean birth as their request may be influenced by the father’s FOC and view that CS is a safer birth option. An intervention that includes the couple may be appropriate in some cases.

## 5. Recommendations for Future Research

This review has highlighted a number of areas for further research. Paternal FOC as a construct is poorly defined and future research needs to distinguish between normal levels of fear surrounding childbirth and severe levels of FOC/tokophobia or posttraumatic stress disorders that may be experienced by men. The development of valid, reliable, acceptable and culturally appropriate FOC screening tools specific for men that distinguish between diagnosis are important in order to identify men requiring additional support and to develop and evaluate appropriate psychological interventions. Further investigation is also recommended to identify predicting factors for father’s development of FOC, knowledge of which would support those providing antenatal care to target timely assessment, and access to treatment and resources to those most in need of support.

The distinction between primary and secondary FOC in the literature is poor as limited research has explored men’s experiences of developing FOC following a previous traumatic birth, miscarriage, stillbirth or neonatal death. Further research is required to establish whether these men with FOC have additional needs to that of men experiencing FOC in other circumstances and findings may be used to inform the development of appropriate care pathways. Currently, in many countries, men are not routinely asked about their fears surrounding childbirth and further co-designed research will need to establish with men the most appropriate and acceptable screening method. Identification of men with FOC will have little success without the availability of evidence-based treatment interventions.

## 6. Limitations

Data extraction from the primary studies of this review was complex due to the wide range of variables studied across the multiple reports and was further impacted upon by challenges of discerning pathological FOC from what might be considered normal fears in relation to childbirth. The findings of this review should be interpreted in light of the methodological limitations within the included studies. These include small sample sizes [1,9,16,42] sample bias, and high attrition rates [43,48]. Eight of the studies included only native-speaking participants, therefore excluding participants of other nationalities [4,11,13,14,17,40,46,48]. Including participants from a variety of cultures and nationalities may achieve greater sample diversity and recognises that giving birth in a foreign country may be a risk factor for developing FOC [73]. Research was predominately represented from Western societies which have medically advanced health care and lower obstetric risks than in other parts of the world. Paternal childbirth fears are likely to be experienced differently in countries that have more medical risk associated with childbirth limiting this reviews generalisability [74]. Language bias is present in this review as studies were restricted to those written in the English language only.

## 7. Conclusions

This review utilised a systematic and rigorous approach in identifying and examining the current available literature reporting on fathers’ experiences of FOC. Non-disclosure of childbirth-related fears is a significant issue among expectant fathers for fear that others would not understand or be dismissive of their fears. Men require opportunities to express their childbirth-related fears and anxieties and to have their fears validated by health professionals. Promoting fathers’ childbirth-related self-efficacy through antenatal care and education is recommended to improve fathers’ confidence and feelings of preparedness for childbirth. Further research is warranted to explore risk factors for FOC, secondary FOC, outcomes of FOC, screening methods and evidence-based interventions and care pathways for treatment and referral for paternal FOC. This review recommends that a family centred perinatal mental health model is required which focuses on maximising the wellbeing of each parent to facilitate family wellbeing.

## Figures and Tables

**Figure 1 ijerph-18-01231-f001:**
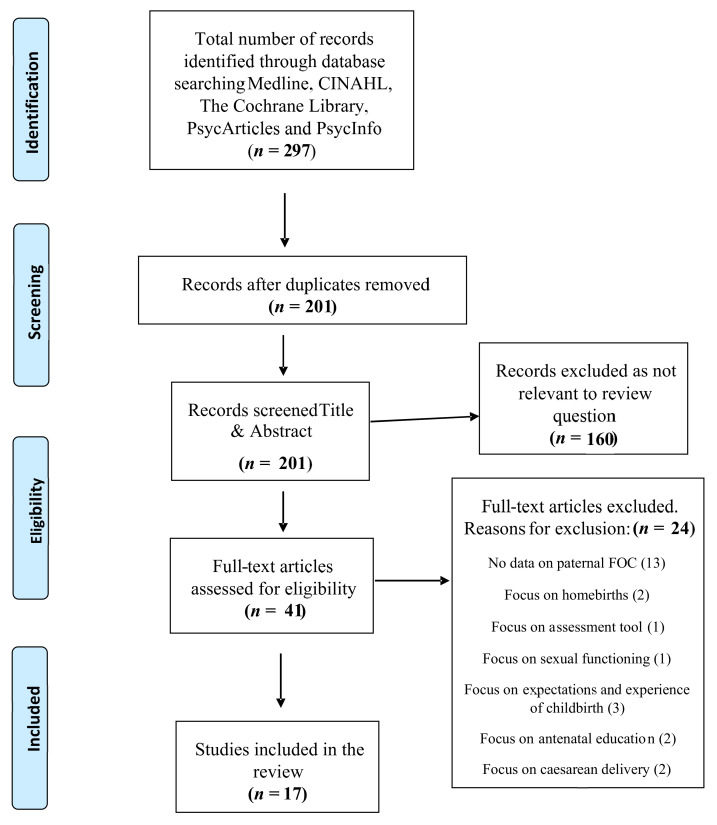
Prisma flow diagram. FOB = Fear of Childbirth.

**Figure 2 ijerph-18-01231-f002:**
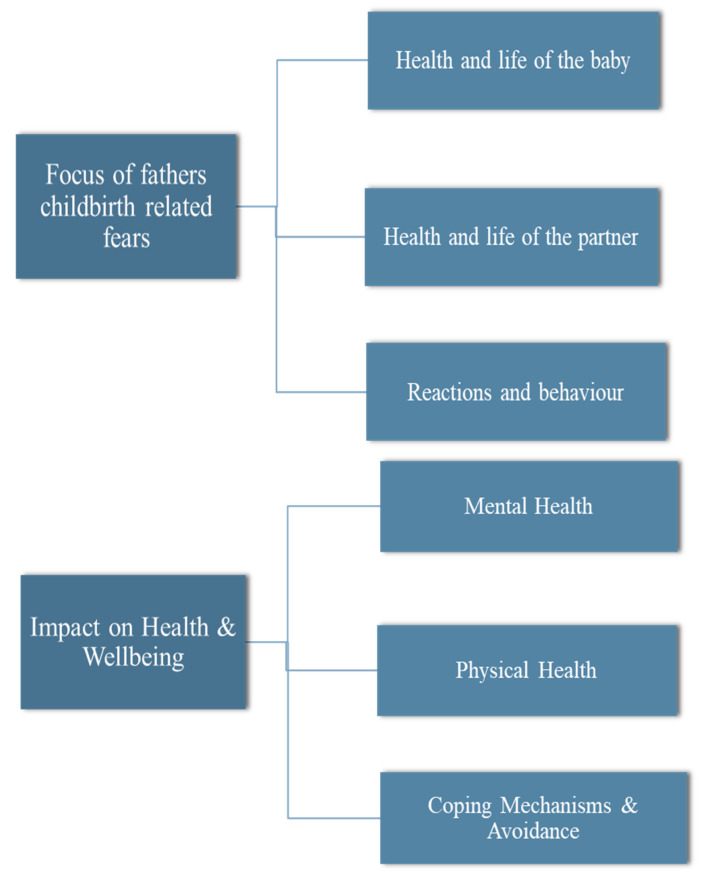
Themes and subthemes.

**Table 1 ijerph-18-01231-t001:** Inclusion and exclusion criteria.

Inclusion Criteria	Exclusion Criteria
Quantitative, qualitative and mixed-methods studies.	Studies examining fathers with other co-morbidities or mental health diagnoses.
Studies relating to the experience of FOC * among fathers. Studies with data relating to FOC among fathers. Studies examining FOC among couples with extractable data relating to fathers. Studies among both first-time fathers and fathers with previous children. Studies published in English.	Studies relating to FOC where a diagnosis of foetal abnormality had been confirmed in pregnancy. Studies published in a language other than English.

* FOC; Fear of Childbirth.

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
