# Peer review of "The Paternal Experience of Fear of Childbirth: An Integrative Review"

_ijerph, 2021, doi:10.3390/ijerph18031231_

Round 1

Reviewer 1 Report

This is a systematic review focused on tokophobia in men, a topic on which there is a paucity of research. The rationale of this review is well explained and justified. The review followed the PRISMA recommendations. Eligible studies were assessed by CCAT. The included studies are well characterized. The qualitative analyses and the identification of main themes and subthemes are sound and congruent. The results, correctly discussed, are regarded as preliminary, and the limitations have been acknowledged. Some recommendations are suggested with potential clinical benefit among men suffering tokophobia. The topic addressed is worthy of investigation. The Methodology is correctly designed and described with enough detail to understand the procedures adopted. The sample of studies included is adequately described for the purposes of this work. The conclusion is in agreement with the body of knowledge regarding paternal tokophobia. In my opinion this review deserves to be published once the following aspects have been improved:

1) Please elaborate on the distinction between fear of childbirth (not always pathological) and tokophobia (pathological), and address the possible limitations caused by the equation of these terms.
2) Given the possibility of a clinical presentation of tokophobia as part of a depressive syndrome in women, please discuss if this prevalent condition can lead to tokophobia also in men.
3) It would be of interest for the readers and of potential clinical relevance to deepen the question of if there are differential profiles in men and women of risk factors (including personality features) and clinical presentation of tokophobia. Also, it would be very interesting to pose (and elaborate) the question of if these putative differences by sex may translate into different therapeutic approaches.

Author Response

 Revision Sheet

Manuscript title: The paternal experience of fear of childbirth: an integrative review

Referee 1

Point raised by referee (please summarise)

Response by author (briefly explain)

Location in text:

Page and paragraph reference

This is a systematic review focused on tokophobia in men, a topic on which there is a paucity of research. The rationale of this review is well explained and justified. The review followed the PRISMA recommendations. Eligible studies were assessed by CCAT. The included studies are well characterized. The qualitative analyses and the identification of main themes and subthemes are sound and congruent. The results, correctly discussed, are regarded as preliminary, and the limitations have been acknowledged. Some recommendations are suggested with potential clinical benefit among men suffering tokophobia. The topic addressed is worthy of investigation. The Methodology is correctly designed and described with enough detail to understand the procedures adopted. The sample of studies included is adequately described for the purposes of this work. The conclusion is in agreement with the body of knowledge regarding paternal tokophobia. In my opinion this review deserves to be published once the following aspects have been improved:

Thank you for this very valuable, constructive critique of our paper. We have attended to your suggestions as outlined below.

Please elaborate on the distinction between fear of childbirth (not always pathological) and tokophobia (pathological), and address the possible limitations caused by the equation of these terms.

Thank you for identifying the confusion that this fear of childbirth Vs tokophobia terminology can bring. Indeed, fear of childbirth exists on a continuum with Tokophobia being the extreme fear and the term tokophobia more frequently used in relation to women. For clarity the term Fear of childbirth (FOC), will be used throughout this paper. The included studies utilise this term so we feel this decision is appropriate.

Lines 55, 58-62

Given the possibility of a clinical presentation of tokophobia as part of a depressive syndrome in women, please discuss if this prevalent condition can lead to tokophobia also in men.

Although it is very probable that men with a diagnosis of depression during the perinatal period are at increased risk of developing FOC and likewise men who experience FOC may be at increased risk of perinatal depression and anxiety.  The selected studies did not specifically aim to examine this link. The paucity of research correlating paternal FOC with perinatal depression and anxiety is highlighted throughout this review as an important factor which requires attention. 

Lines 57, 437-438,

It would be of interest for the readers and of potential clinical relevance to deepen the question of if there are differential profiles in men and women of risk factors (including personality features) and clinical presentation of tokophobia. Also, it would be very interesting to pose (and elaborate) the question of if these putative differences by sex may translate into different therapeutic approaches.   

Thank you for this suggestion which would, we agree deepen the question but is may be beyond the scope of this review. Reference to some of these factors are noted in the discussion section detailed below and in Lines 337 -341 and we have also included this as an area requiring further investigation (Line 516-519)77-382.

However, findings of this review offer only a limited examination of the factors associated with men’s risk of developing FOC outside of broad demographic explanations

337-341, 437-438

Reviewer 2 Report

This is a methodologically sound integrative review following the PRISMA guidelines. I applaud the work that has gone into the paper and the clarity of the presentation of findings.

There is, however, a major concern with this paper and that is your use of 'fear of childbirth' interchangeably with 'tokophobia'. It is far from clear in psychological and mental health terms that the two are indeed interchangeable and it cannot be assumed that the literature on FOB to which you refer at great length is in fact talking about tokophobia. Fear of childbirth in so far as men worry that their baby might die during birth, or their partner, and that they will feel helpless seeing their partner in pain, is, I would suggest, sufficiently widespread to be considered 'normal' (but nonetheless distressing) rather than 'pathological'. The Limitations section of your paper needs to revisit the difficulty in differentiating a pathological level of fear from a normal level, and the Discussion section needs to be clear that while fearful fathers are, of course, deserving of support and compassion from midwifery and medical staff, there is a need to identify and target resources at those who have a disabling fear of childbirth. 

The paper would benefit from being considerably shorter, especially in relation to the Discussion section where the literature cited ranges far beyond the focus of the integrative review itself. The PRISMA guidance suggests that the key areas of the review are the Methods and Results sections followed by a brief Discussion providing 'a general interpretation of the results in the context of other evidence'.

The paper also requires some additional editing as there is a recurring problem with misuse of the apostrophe (some instances: lines 13; 33; 89; 95; 108; 116; 478; 504 et al.); random use of capitals (e.g. lines 259; 325) and incorrect punctuation in relation to 'however' (e.g. line 490).

In order to be publishable, this paper needs to acknowledge the difficulty in defining and distinguishing between FOB and tokophobia. The Discussion needs to summarise the results of the review briefly (as they have already been described at length in the preceding section of the paper); provide a far more focused and succinct account of how the literature bears out the findings; be clear about the strengths and limitations of the review, and briefly suggest how the study might impact antenatal clinical care and point the way to future research.

Author Response

Revision Sheet

Manuscript title: The paternal experience of fear of childbirth: an integrative review

Referee 2

Point raised by referee (please summarise)

Response by author (briefly explain)

Location in text:

Page and paragraph reference

This is a methodologically sound integrative review following the PRISMA guidelines. I applaud the work that has gone into the paper and the clarity of the presentation of findings.

Thank for this acknowledgement of the work undertaken in this review.

There is, however, a major concern with this paper and that is your use of 'fear of childbirth' interchangeably with 'tokophobia'. It is far from clear in psychological and mental health terms that the two are indeed interchangeable and it cannot be assumed that the literature on FOB to which you refer at great length is in fact talking about tokophobia. Fear of childbirth in so far as men worry that their baby might die during birth, or their partner, and that they will feel helpless seeing their partner in pain, is, I would suggest, sufficiently widespread to be considered 'normal' (but nonetheless distressing) rather than 'pathological'. The Limitations section of your paper needs to revisit the difficulty in differentiating a pathological level of fear from a normal level, and the Discussion section needs to be clear that while fearful fathers are, of course, deserving of support and compassion from midwifery and medical staff, there is a need to identify and target resources at those who have a disabling fear of childbirth.

Thank you for this suggestion and we agree with your comments. To address this, we have amended the paper to focus on Fear of childbirth in men and removed reference to tokophobia. We have differentiated between a pathological fear of childbirth among fathers from the more common worries and concerns experienced by many fathers to be in anticipation of childbirth. This is detailed in the Introduction section: Lines 55, 58-62. We have acknowledged this as suggested in the Limitations section, Lines 431-434. The two terms are often used interchangeably within the literature, however we acknowledge it may cause confusion and lack of clarity.

Lines 55, 58-62

Lines 431-434

The paper would benefit from being considerably shorter, especially in relation to the Discussion section where the literature cited ranges far beyond the focus of the integrative review itself. The PRISMA guidance suggests that the key areas of the review are the Methods and Results sections followed by a brief Discussion providing 'a general interpretation of the results in the context of other evidence'.

We have reviewed the contents of the paper with a particular focus on the discussion section and the paper is now considerably shorter.

The paper also requires some additional editing as there is a recurring problem with misuse of the apostrophe (some instances: lines 13; 33; 89; 95; 108; 116; 478; 504 et al.); random use of capitals (e.g. lines 259; 325) and incorrect punctuation in relation to 'however' (e.g. line 490).

This paper has benefited from careful and repeated proofreading with associated editing and we hope that we have captured all of our previous errors 

In order to be publishable, this paper needs to acknowledge the difficulty in defining and distinguishing between FOB and tokophobia. The Discussion needs to summarise the results of the review briefly (as they have already been described at length in the preceding section of the paper); provide a far more focused and succinct account of how the literature bears out the findings; be clear about the strengths and limitations of the review, and briefly suggest how the study might impact antenatal clinical care and point the way to future research.

Thank you for your valuable guidance in preparing the paper for publication. The difficulties associated with differentiating FOB and tokophobia have been discussed and we have removed reference to tokophobia. We have reviewed and reduced the discussion section, removed repetition, identified the Limitations more succinctly, made some suggestions for further research and highlighted the importance of antenatal care and education for fathers to be in addressing FOC.  

Lines 55, 58-62

Reviewer 3 Report

This review aimed at examining and synthesizing the current body of evidence of research relating to paternal experience of tokophobia. While the topic is interesting, I have concerns about this paper.

The aims and ultimate usefulness of described findings should be clarified and structure and description of the methods should be improved.

Aim(s)

The general aim of this review seems to be rather generally described. In par 2.1 the aim is described as gaining insight and holistic understanding of men's tokophobia or FOB.

However, a clearer aim/focus of the review is to be preferred. Moreover, although several indications or reasons are given throughout the introduction why FOB can be harmful for both women (e.g. increased risk of cesarean section) and the males themselves (e.g. negative impact on parenting ability), par. 1.3. (entitled 'Rational for undertaking this review') does not describe these clear reasons systematically, nor why a review to summarize all the evidence is needed and what this review could contribute to solving to FOB related problems for fathers, mothers and their children.

Ad 1. Introduction

In lines 73-77 the criticisms on the W-DEQ are described extensively, especially when using this instrument in men. However, it is not quite clear why the author paid so much attention to the weaknesses of this instrument in this particular paragraph. Why did the author not describe other instruments? And what is the relation of this topic with the main aim? Could this be explained better? 

Ad 2.2. Search strategy

The search strategy is rather old and should be updated. Inclusion and exclusion criteria have not been described; in Figure 1 it has been described that 160 papers were not relevant for the review question. However, the research question is very global, implying that the reasons for exclusion and inclusion are not clear at all to the readers.

Ad 2.4

It was not explained why only 17 out of 41 papers were included in the review: which quality criteria had to be met exactly? And why were criteria considered to be met unsufficiently in the papers that were not included?

Table 1 should have a title at the top of the table, which exactly explains what is the contents of this table. Moreover the aims of the studies included in this table as far as they are related to the aim of the review should have been described very clearly. Now, it is not clear which messages the author wants to communicate via this table. Moreover, Table 1 could have been better readable in landscape format.

ad 3.2 and the other paragraphs in which identified themes are described.

The author indicated what is the first, second and third theme that emerged from literature. Apparently, one of the aims of this review was to identify themes. This should be included in the aim of this review. 

Minor comments:

  • Although FOB and tokophobia are synonyms according to the author, both expressions are used interchangeably or sometimes even simulataneously. Why not just use one of these after explaing that these words mean exactly the same?

Author Response

Revision Sheet

Manuscript title: The paternal experience of fear of childbirth: an integrative review

Referee 3

Point raised by referee (please summarise)

Response by author (briefly explain)

Location in text:

Page and paragraph reference

This review aimed at examining and synthesizing the current body of evidence of research relating to paternal experience of tokophobia. While the topic is interesting, I have concerns about this paper.

The aims and ultimate usefulness of described findings should be clarified and structure and description of the methods should be improved.

Thank you for your guidance in relation to the review.  We have revised the aim and we have improved the structure and description of the methods section.

The general aim of this review seems to be rather generally described. In par 2.1 the aim is described as gaining insight and holistic understanding of men's tokophobia or FOB.

However, a clearer aim/focus of the review is to be preferred. Moreover, although several indications or reasons are given throughout the introduction why FOB can be harmful for both women (e.g. increased risk of caesarean caesarean section) and the males themselves (e.g. negative impact on parenting ability), par. 1.3. (entitled 'Rational for undertaking this review') does not describe these clear reasons systematically, nor why a review to summarize all the evidence is needed and what this review could contribute to solving to FOB related problems for fathers, mothers and their children.

 The aim of the review has been revised taking into account the feedback from reviewer.

106-112

Ad 1. Introduction

In lines 73-77 the criticisms on the W-DEQ are described extensively, especially when using this instrument in men. However, it is not quite clear why the author paid so much attention to the weaknesses of this instrument in this particular paragraph. Why did the author not describe other instruments? And what is the relation of this topic with the main aim? Could this be explained better?

This has been revised with a briefer comment on the W-DEQ tool. Detail of this tool was included as it is most frequently used tool to screen for FOB  

Lines 64-71

Ad 2.2. Search strategy

The search strategy is rather old and should be updated. Inclusion and exclusion criteria have not been described; in Figure 1 it has been described that 160 papers were not relevant for the review question. However, the research question is very global, implying that the reasons for exclusion and inclusion are not clear at all to the readers.

Due to time constraints, it has not been possible to conduct a comprehensive search for relevant literature on this subject from Jan 2020 to Nov 2020. However, a quick search of the databases did not reveal any further publications in that time period. Given that this was not a detailed search, we feel it would be inappropriate to revise the timelines of the search strategy. 

Inclusion and exclusion criteria have been clearly delineated in Table 1.

The review aim has been revised.

Line 130

Lines 106-109

Ad 2.4

It was not explained why only 17 out of 41 papers were included in the review: which quality criteria had to be met exactly? And why were criteria considered to be met insufficiently in the papers that were not included?

The inclusion of clearly delineated inclusion and exclusion criteria will help the reader to determine why 24 papers were excluded from the final review and further detail has been provided in the Prisma flow diagram as to why papers were excluded text indicating that the papers were excluded from the review because they were not relevant to the research question 

Table 1 should have a title at the top of the table, which exactly explains what is the contents of this table. Moreover, the aims of the studies included in this table as far as they are related to the aim of the review should have been described very clearly. Now, it is not clear which messages the author wants to communicate via this table. Moreover, Table 1 could have been better readable in landscape format.

 The title is present on the table and it is presented in a landscape format. A clear description of this tables contents is made in lines 160-164. Only studies that met inclusion criteria were included. 

ad 3.2 and the other paragraphs in which identified themes are described.

The author indicated what is the first, second and third theme that emerged from literature. Apparently, one of the aims of this review was to identify themes. This should be included in the aim of this review.

The review question has been clarified, as noted earlier. Thematic analysis of the data was used to synthesise the findings as referred to in the review question and therefore was not an aim of the review but a means of synthesis. 

Minor comments:

Although FOC and tokophobia are synonyms according to the author, both expressions are used interchangeably or sometimes even simultaneously. Why not just use one of these after explaining that these words mean exactly the same?

Thank you for identifying the confusion that this fear of childbirth Vs tokophobia terminology can bring. Indeed, fear of childbirth exists on a continuum with Tokophobia being the extreme fear and the term tokophobia more frequently used in relation to women. For clarity the term Fear of childbirth (FOC), will be used throughout this paper. The included studies utilise this term so we feel this decision is appropriate.

Round 2

Reviewer 2 Report

Thank you for responding appropriately to my review. Your reconsideration of the terms 'FOC' and 'tokophobia' has strengthened the study. Likewise, the Discussion section is now more succinct and focused.

My only remaining criticism relates to punctuation and a few instances where a single subject is followed by a plural verb and vice versa. Please amend as follows:

line 45: a father's

line 54: affect

line 54: [11];

line 59: affects

line 80: a father's

line 81: his partner's

line 94: A father's

line 95: a father's

line 106: health professionals'

line 152: studies were

line 154: were compared

line 160: four qualitative studies mentioned but only three listed in the sentence beginning 'Qualitative study designs consisted of....'

line 164 was not clearly

line 175: presented secondary analysis

line 197: with FOC

line 209: predominant fear was

line 246: preoccupation in which

line 247: the fear brought with it

line 313: receiving support were reported

line 351: for fear that doing so would detract

line 402: of these elements helps strengthen

line 410: fears was the most dominant

line 420: their request may be

line 421: the fathers' FOC

line 431: between diagnoses

line 433: for fathers'

line 460: this review's